# Antenatal, Intrapartum and Postpartum Interventions for Preventing Postpartum Urinary and Faecal Incontinence: An Umbrella Overview of Cochrane Systematic Reviews

**DOI:** 10.3390/jcm12186037

**Published:** 2023-09-18

**Authors:** Juliette Sananès, Sophie Pire, Anis Feki, Michel Boulvain, Daniel L. Faltin

**Affiliations:** 1Department of Obstetrics and Gynecology, HFR—Cantonal Hospital of Fribourg, 1752 Villars-sur-Glânes, Switzerland; 2Centre de Périnéologie Dianuro Geneva, 1227 Carouge, Switzerland

**Keywords:** postpartum incontinence, postpartum faecal incontinence, postpartum urinary incontinence

## Abstract

Post-partum, women can suffer from urinary and faecal incontinence. It is important to assess interventions to prevent this problem. Cochrane systematic reviews summarize the data available from systematic reviews of randomized trials assessing interventions. We conducted an umbrella overview of Cochrane systematic reviews encompassing antenatal, intrapartum and postpartum interventions for preventing postpartum urinary and faecal incontinence. We searched the Cochrane Database of Systematic Reviews on the 9 May 2023. Results: Our search identified nine Cochrane reviews providing results. Data for urinary and faecal incontinence were available from 77 (72%) trials and included 51,113 women. The reviews assessed antenatal digital perineal massage, pelvic floor muscle training, techniques for repairing anal sphincter tears, routine use of episiotomy, use of endoanal ultrasound prior to repairing perineal tears, caesarean versus vaginal delivery (overall, for breech and for twins), and vaginal delivery with forceps or vacuum. Only the use of a vacuum instead of forceps if an assisted vaginal delivery is needed, the use of an endo-anal ultrasound prior to repairing perineal tears and postpartum pelvic floor muscle training suggest a reduction in postpartum incontinence. Due to the small number of relevant reviews, a consequence of the relatively small number of primary studies, the effect of almost all the tested interventions was found to be imprecise.

## 1. Introduction

### What Is the Issue

The prevalence of stress urinary incontinence in the first three months following delivery is approximately 8% for the primiparous patients and about 20% for the multiparous women [1].

Faecal incontinence at three months postpartum may be 19% to 46% for flatus and 2.4% to 8.0% for the involuntary loss of formed stool [2]. In the longer term, these rates seem to persist, with about 31% of primiparous women reporting involuntary loss of flatus at six and twelve years after delivery and 9% to 12% reporting loss of formed stool [3]. One systematic review suggested that the etiological factor most strongly associated with postpartum faecal incontinence is a third- or fourth-degree rupture of the external anal sphincter [4].

Managing incontinence after pregnancy is not only important for the individuals themselves but can also have considerable costs to individuals and for healthcare systems.

Due to the fact that there are various risk factors for urinary or faecal incontinence, different interventions can help to prevent incontinence in postpartum. This summary provides an overview of the findings from Cochrane reviews on interventions during pregnancy and childbirth aimed at preventing urinary or faecal incontinence in the postpartum period.

## 2. Materials and Methods

### 2.1. Objectives

The aim of this summary was to provide an overview of the evidence found in Cochrane reviews concerning the impact of interventions during pregnancy, labour and the postpartum period on the prevention of urinary and faecal incontinence.

### 2.2. Criteria for Inclusion

We incorporated Cochrane reviews that examined antenatal, intrapartum or postpartum interventions with outcomes reporting on postpartum urinary or faecal incontinence. The postpartum period under consideration was immediately after the delivery and up to 3 years.

We listed the reviews that specified urinary or faecal incontinence as an outcome but had no data reported (see Appendix B: Reviews awaiting further classification). These reviews will be taken into account in future updates of the overview if additional data becomes available.

### 2.3. Participants

We examined reviews that evaluated interventions for pregnant women during the antenatal (that is to say during the pregnancy), childbirth or postpartum (up to 3 years after the delivery) phases.

### 2.4. Interventions

We examined all interventions employed during the antenatal, peripartum and postpartum period compared with an alternative intervention.

### 2.5. Outcomes of Interest

Urinary and faecal incontinence were the main outcomes. When urinary and faecal incontinence were assessed after more than one period of time, the last assessment was considered.

### 2.6. Search Strategy

We conducted a search in the Cochrane Database of Systematic Reviews, employing the terms “incontinence or perineum or episiotomy” AND “pregnancy or birth or delivery or partum or labour or natal” in the title, abstract or keywords. Word variations have been searched (last searched on 9 May 2023). We conducted our search with “all text”, without imposing restrictions on “title, abstract, or keywords”. We imposed no limitations regarding language or publication date, and we did not explore any additional databases.

### 2.7. Data Collection and Analysis

#### 2.7.1. Selection of Reviews

Two overview authors independently assessed all potential reviews we identified. We resolved any disagreement through discussion, or if required, we consulted a third author.

#### 2.7.2. Data Extraction and Management

Two authors independently extracted data from the reviews using a pre-defined data extraction form. We resolved discrepancies through discussion.

We extracted information on the following:

Review characteristics:review title and authors;date that the review was last assessed as up-to-date;number of included trials;number of participants in the trials and their characteristics;quality of the included trials (as reported by the review authors);interventions pertinent to this overview;all predetermined outcomes pertinent to this overview.

Statistical summaries:number of studies and participants contributing data relevant to this overview;the summary intervention effects: risk ratios (RR) or odds ratios (OR) with 95% confidence intervals (CI);data needed to evaluate the quality of evidence regarding the intervention’s impact.

#### 2.7.3. Assessment of Methodological Quality of Included Reviews

##### Quality of Included Reviews

Two overview authors independently assessed the quality of the included reviews; another overview author verified this assessment. We resolved differences through discussion.

##### Methodological Quality

We assessed the methodological quality of each review using the AMSTAR 2 (A Measurement Tool to Assess Reviews) tool. AMSTAR 2 evaluates the methods used and assesses the extent to which review methods exhibit impartiality. The criteria are assessed as either yes, no, partially yes, or not applicable. These criteria are summarized in Appendix C.

A review that has all 16 criteria was considered of high quality. The ratings for overall confidence in the results of the reviews are the following:High

No or one non-critical weakness: the systematic review provides an accurate and comprehensive summary of the results;

Moderate

More than one non-critical weakness: the systematic review has more than one weakness but no critical flaws. It may provide an accurate summary of the results;

Low

One critical flaw with or without non-critical weaknesses: the review has a critical flaw and may not provide an accurate and comprehensive summary;

Critically low

More than one critical flaw with or without non-critical weaknesses: the review has more than one critical flaw and should not be relied on to provide an accurate and comprehensive summary of the available studies.

##### Risk of Bias

To assess the systematic reviews, we used the ROBIS (Risk OF Bias In Systematic reviews) tool. It considers four domains:Study eligibility criteria;Identification and selection of studies;Data collection and study appraisal;Synthesis and findings.

Once the assessors have appraised these queries (low, high or unclear), they weigh the review’s collective risk of bias.

##### Quality of Included Studies within Reviews

We did not assess the quality of included studies within reviews but reported study quality according to the review authors’ assessment.

##### Quality of Evidence in Included Reviews

We evaluated the robustness of evidence pertaining to our key outcome (urinary and faecal incontinence) using the GRADE approach. We reported the quality of evidence based on the assessments made by the researchers conducting the reviews, using ‘Summary of findings’ tables when available. The GRADE system examines the following aspects:

Study limitations: internal validity of the evidence;Inconsistency: discrepancy or fluctuation in the effect measurements across studies;Indirectness: extent of disparities among populations;Imprecision (random error): the level of confidence in the effect estimate to make informed decisions;Publication bias: the degree to which studies with certain results are selectively published.

The GRADE system scores the quality of the evidence as follows:

High: additional research is highly improbable to alter our level of confidence in the effect estimate;Moderate: additional studies are likely to substantially influence our level of confidence in the effect estimate and could potentially lead to revisions in the estimate;Low: further research is highly likely to significantly impact our confidence in the effect estimate and is likely to change the estimate;Very low: the effect estimate is highly uncertain.

##### Data Synthesis

We summarized the main results of the included reviews by organizing by intervention topic.Effective interventions: the review provided strong evidence supporting their effectiveness;Potentially effective interventions: the review provided reasonable evidence supporting their effectiveness;Ineffective interventions: the review provided strong evidence showing they are not effective;Probably ineffective interventions: the review provided reasonable evidence suggesting they are not effective;No conclusion possible: the review had limited or very uncertain evidence.

We determined the category for each intervention by assessing the quality of evidence pertaining to the management of urinary or faecal incontinence.

## 3. Results

Our search of the Cochrane Database of Systematic Reviews identified 72 reviews that could be potentially relevant.

After reviewing the titles and abstracts, we eliminated 26 reviews and conducted an in-depth analysis of 46 protocols and reviews by examining their full texts.

We excluded 28 reviews because they did not include urinary or faecal incontinence as an outcome (Table 1: Characteristics of excluded reviews).

We enumerated 13 reviews that mentioned urinary or faecal incontinence as an outcome, but the trials included in these reviews did not provide any data. We will consider these reviews for possible incorporation in future revisions of the overview if additional data becomes available (see Appendix A: reviews awaiting further classification).

Two reviews, that specify urinary or faecal incontinence as outcomes, are currently awaiting classification because they have not yet identified eligible trials (Appendix B).

Nine reviews were incorporated into this overview.

See Figure 1. Review flow diagram.

Description of included reviews.

About the nine included reviews:
Three evaluated vaginal delivery versus caesarean delivery:
○In a global population of pregnant women [5];○In women with breech presentation [6];○In women with twin pregnancy [7].One compared instruments (forceps and vacuum) for assisted delivery [8].One evaluated the effects of antenatal perineal massage [9].One evaluated the effects of antenatal or postnatal pelvic floor muscle training [10].One compared performing episiotomy only if needed versus routine episiotomy [11].Two evaluated the repair of perineal tears:○One compared the use of endoanal ultrasound prior to repairing perineal tears versus routine care [12];○One compared two repairing techniques for obstetrics anal sphincter injuries: overlap versus end-to-end [13].

The count of randomized controlled trials (RCT) in the 9 reviews varied from 1 [5,12] to 46 [10]. One review included twenty non-randomized but controlled studies [5].

The number of women in each study varied, ranging from 588 [13] to 31,698 [5]. In the complete count, we had 127 trials involving over 64,401 women.

The reviews conducted searches at different times, spanning from July 2009 to August 2019. See Table 2, characteristics of included reviews.

### 3.1. Methodological Quality of Included Reviews

According to AMSTAR criteria:All reviews indicated the components of PICO;All reviews predetermined their study design;All reviews pre-specified which study design will be included, but only four explain why they choose this design;All reviews used a comprehensive search strategy;All reviews reported that study selection and data extraction were conducted independently by different people;All reviews presented lists of studies that were excluded;All reviews provided detailed characteristics of the studies that were included;All reviews used an appropriate and satisfactory method for evaluating the risk of bias in the included studies;Six reviews included the sources of funding for the studies [6,7,10,11,12,13];Eight reviews used appropriate methods for the statistical combination of results and assessment of the potential impact of risk of bias on the results of the meta-analysis. The last review conducted no meta-analysis because only one trial was included [12];All the reviews took into account the risk of bias in individual studies when interpreting the results;All the reviews provided a satisfactory explanation for the heterogeneity observed in the results;None of the reviews carried out an adequate investigation of publication bias;Only one review did not report any potential sources of conflict of interest [11].

See Table 3: AMSTAR assessments for included reviews, for further details.

With the ROBIS tool, it was determined that all reviews exhibited a low risk of bias. See Table 4 ROBIS assessments for included reviews, for further details.

### 3.2. Effects of Interventions

Below, we have provided a summary of the main findings from the reviews we included. These findings are categorized by intervention topic, considering the quality of evidence for their impact on urinary or faecal incontinence.

For additional information and the ‘Summary of findings’ table for the outcomes of interest, see Table 5.

No conclusion possible: low-quality evidence.

The review on selective or routine episiotomy [11] showed no evidence of a difference for the prevention of urinary incontinence (RR 0.98, 95% CI 0.67 to 1.44; three trials; 1107 women).

The review on techniques for repairing anal sphincter injuries [13] showed no evidence of a difference between end-to-end repair and overlap repair for the prevention of faecal incontinence at 6 weeks (RR 0.65, 95% CI 0.20 to 2.07, 1 trial, 63 women), 3 months (RR 0.84 95% CI 0.06 to 12.73, 2 trials, 201 women), 6 months (RR 0.48 95% CI 0.02 to 12.89, 2 trials, 205 women), 12 months (RR 0.37 95% CI 0.03 to 4.68, 3 trials, 256 women), 24 months (RR 0.88, 95% CI 0.32 to 2.41, 1 trial, 95 women) and 36 months (RR 1.01, 95% CI 0.34 to 2.98, 1 trial, 68 women).

The review comparing planned caesarean or planned vaginal delivery for women with a breech presentation [6] showed a decrease in urinary incontinence at 3 months (RR 0.62 95% CI 0.41 to 0.93, 1 trial with 1595 women) but no difference was detected for faecal incontinence at 3 months (RR 0.54 95% CI 0.18 to 1.62, 1 study, 1226 women) and no evidence of a difference was detected at 2 years for both urinary incontinence (RR 1.14, 95% CI 0.81 to 1.61) and faecal incontinence (RR 1.11 95% CI 0.47 to 2.58, 1 study, 917 women).

### 3.3. Probably Ineffective Interventions: Moderate-Quality Evidence

The review on antenatal digital perineal massage [9] did not show a difference in urinary incontinence (RR 0.90 95% CI 0.74 to 1.08) or faecal incontinence (RR 0.70 95% CI 0.27 to 1.80, four trials, 2497 women).

The review comparing planned caesarean and a planned vaginal delivery for women with a twin pregnancy [7] did not show a difference in urinary (RR 0.87 95% CI 0.64 to 1.18) or faecal incontinence (RR 1.02 95% CI 0.69 to 1.85, 2 trials, 2864 women).

The review comparing caesarean delivery and vaginal delivery [5] did not find a benefit for faecal incontinence (OR 0.93 95% CI 0.77 to 1.13, 20 observational studies, 1 trial, 31,698 women).

### 3.4. Probably Effective Interventions: Moderate-Quality Evidence

Performing an endo-anal ultrasound after childbirth and prior to repairing any perineal tears [12] probably reduced severe faecal incontinence before 6 months postpartum (RR 0.48 95% CI 0.24 to 0.97, 1 trial, 752 women) and from 6 to 12 months post-partum (RR 0.38; 95% CI 0.20–0.72). However, it seems to not be effective for non-severe faecal incontinence before and after 6 months (RR 0.95, 95% CI 0.73–1.22).

Pelvic floor muscle training [10] may reduce urinary incontinence in early and mid-postnatal periods (RR 0.38 95% CI 0.17 to 0.83, 5 trials, 439 women for early postnatal period; RR 0.71 95% CI 0.54 to 0.95, 5 trials, 673 women for mid-postnatal period) but no evidence of a difference was provided in the late postnatal period (RR 1.20 95% CI 0.65 to 2.21, 1 trial, 44 women for late postnatal period).

Using a vacuum instead of forceps in the case of assisted vaginal delivery [8] seems to cause less faecal incontinence (RR 0.56 95% CI 0.61 to 0.84, 1 trial, 130 women).

## 4. Discussion

### 4.1. Summary of Main Results

Nine Cochrane reviews were included in this overview, encompassing 67 randomized controlled trials (21,355) and 20 non-randomized controlled studies (29,758).

Probably effective interventions: a possible reduction in faecal incontinence was found for the following:When performing an endo-anal ultrasound after childbirth and prior to repairing any perineal tears (moderate-quality evidence);When using a vacuum instead of forceps for instrumented vaginal deliveries (moderate-quality evidence);When pelvic floor muscle training is performed postpartum (moderate-quality evidence).

Probably ineffective interventions:

No evidence of a difference in urinary or faecal incontinence when antenatal perineal massage is performed (moderate-quality evidence), a planned caesarean delivery instead of a planned vaginal delivery for women (overall, for twins and for breech) (moderate-quality evidence). There was a short-term decrease in faecal incontinence in the case of a planned caesarean for breech, but no effect in the longer term.

No conclusion possible:

No clear difference in preventing urinary or faecal incontinence when performing a routine or selective episiotomy or when using the overlap or end-to-end repair for obstetrics anal sphincter injuries (low quality evidence).

### 4.2. Overall Completeness and Applicability of Evidence

This overview presents a summary of Cochrane reviews that have evaluated various antenatal, intrapartum and postpartum interventions and their effects on urinary and faecal incontinence.

We included a total of nine reviews, which reported data on the primary outcome and urinary and faecal incontinence. Additionally, we found seven protocols that have already specified urinary or faecal incontinence as secondary outcomes. These protocols will be taken into account for potential integration in upcoming revisions of the overview once they are published as full reviews. These protocols are designed to examine interventions related to urinary and faecal incontinence including perineal techniques during the second stage of labour for reducing perineal trauma, planned elective repeat caesarean section versus planned vaginal birth for women with a previous caesarean birth, surgical repair of spontaneous perineal tears that occur during childbirth versus no intervention, position in the second stage of labour for women without epidural analgesia, symphysiotomy for feto-pelvic disproportion, discontinuation of epidural analgesia late in labour and hyaluronidase for reducing perineal trauma. See Appendix A, Ongoing reviews.

We could not involve two more reviews that assessed urinary or faecal incontinence as the primary or secondary outcome, because no trial met the inclusion criteria (despite the fact that they recognized the possible influence of the interventions on incontinence by including it as an outcome in their review).

We have summarized the main conclusions of these reviews in Appendix B (Reviews awaiting further classification). These reviews will be reconsidered for inclusion in future updates of the overview.

### 4.3. Quality of the Evidence

Each of the reviews evaluated the potential bias in the randomized trials they included.

Four out of the nine reviews employed the GRADE approach to appraise the quality of evidence regarding their review outcomes [6,7,10,11]. For the remaining reviews, we employed the GRADE score to assess the quality of evidence, taking into account the study constraints (risk of bias) as indicated by the authors of the reviews. Regarding the primary outcome, urinary and faecal incontinence, the quality of the evidence ranged from low to moderate. The low quality frequently came from inaccuracies arising from limited sample sizes, a scarcity of occurrences and expansive confidence intervals.

Because our overview’s data were derived from longer-term follow-up antenatal, intrapartum or postpartum interventions, we meticulously examined potential biases associated with attrition (where some women were lost to follow-up, possibly leading to differences from those who were followed-up).

Table 5 displays the summary of findings regarding the quality of evidence for our outcomes.

### 4.4. Potential Biases in the Overview Procedure

When studies talked about urinary or faecal incontinence as an outcome, there was little data reported by the patients. This lack of date did not allow us to have complete and detailed information about it.

The subjective nature of the outcomes (self-questionnaires) may limit the quality of the assessment. Symptoms reported by women may be seen, however, as more important compared to any objective measurement (e.g., sphincter tone).

Moreover, all reviews have a low quality of evidence according to the GRADE score.

We only searched in the Cochrane Database of Systematic Reviews. We only chose this one because this is a highly demanding database (see [1]). Using only one database could be a limitation for our review, but the Cochrane Database has rigorous methodology in order to limit the risk of bias guaranteeing a reliable understanding of the evidence [14]. However, it is clear that the heterogeneity of the interventions in some reviews can have a negative impact on the results.

### 4.5. Consensus and Disparities with Other Studies

No other systematic review or overview about interventions for the prevention of urinary or faecal incontinence were identified.

Regarding perineal trauma and urinary and faecal incontinence prevention:Some recommendations included that routine episiotomy does not prevent severe perineal trauma [10];A vaginal delivery with vacuum resulted in less damage to the perineum than the forceps [7];The systematic search of obstetrical anal sphincter injury is necessary after vaginal delivery to adapt the repair [12];Antenatal pelvic floor muscle training seems to be beneficial to prevent incontinence postpartum [9];Caesarean delivery could be beneficial to prevent incontinence post-partum compared to vaginal delivery [4].

Incontinence and prolapse are seen as consequences of vaginal delivery, and a caesarean section may be requested by women to protect the pelvic floor, despite the fact that it is a major surgical procedure with significant complications. Ref. [15] reported data on possible adverse health outcomes for children delivered by caesarean, compared to those delivered vaginally. This includes respiratory illness, atopic conditions, obesity, diabetes and other severe auto-immune diseases. It highlights the need to warn women about these risks and to discuss if caesarean delivery is acceptable only to prevent possible pelvic floor damage. Furthermore, ref. [16] did not show a difference in short-term pelvic floor muscle strength after childbirth between primiparous women who underwent caesarean section or vaginal delivery, which highlights the need to discuss the indication of a caesarean delivery in order to prevent possible pelvic floor damage.

A study [17] suggested that women want to know their risk of developing pelvic floor dysfunction to help make decisions about childbirth. They also report that knowing their risks would motivate them to undertake preventative strategies, such as pelvic floor muscle training. This desire for information is further supported by [17] as part of a “care bundle” that has been shown to reduce the risk of obstetric anal sphincter injury, with no increase in caesarean section rate. This “care bundle” consists in four elements: antenatal information for women about OASIS, use of a manual perineal protection, performing an episiotomy only if needed, performing it mediolaterally with a 60-degree angle at crowning, and a systematic examination of the vagina and ano-rectum after childbirth. Ref. [18] studied women’s experience of the OASIS Care Bundle. Their findings suggest that the four elements of the OASIS Care Bundle are acceptable to women. This highlights the need for antenatal risk assessment and providing women with information on which to make informed choices.

## 5. Authors’ Conclusions

### 5.1. Implications for Practice

This overview summarizes the evidence from Cochrane reviews of randomized controlled trials and non-randomized controlled trials regarding the effects of antenatal, intrapartum and postpartum interventions on urinary and faecal incontinence.

There were no interventions with high-quality evidence of a benefit in this overview.

There were interventions with moderate-quality evidence showing a possible reduction in urinary or faecal incontinence: (1) performing an endo-anal ultrasound after childbirth and prior to repair of any perineal tears, (2) using a vacuum instead of forceps for assisted vaginal deliveries and (3) performing pelvic floor muscle training.

There were interventions with moderate-quality evidence showing no clear difference in urinary or faecal incontinence: (1) performing an antenatal perineal massage and (2) performing a planned caesarean delivery instead of a planned vaginal delivery for women with a twin pregnancy or performing a caesarean delivery instead of a vaginal delivery.

For other interventions examined in this overview, we could not draw any conclusions due to either low-quality evidence or a lack of evidence altogether.

### 5.2. Implications for Research

This overview emphasizes the lack of sufficient evidence to make conclusive statements relative to the impact of different antenatal, intrapartum, and postpartum interventions on urinary and faecal incontinence. It may be attributed to the following:A shortage of primary research exists, with few long-term, follow-up studies on women involving randomized trials of antenatal, intrapartum and postpartum interventions.Insufficient reporting on urinary and faecal incontinence by randomized trials.Insufficient reporting on incontinence postpartum by relevant Cochrane reviews (probably because it was not predefined as an outcome).The absence of a Cochrane review evaluating potentially relevant interventions.

Considering our enhanced comprehension of the various risk factors and etiologies contributing to postpartum urinary and faecal incontinence, there is a pressing demand for conducting long-term, follow- up studies on interventions that target these risk factors. These studies should adhere to rigorous design principles and strive for uniformity in measuring and reporting outcomes related to urinary and faecal incontinence. This approach will enable the pooling of outcome data and, as a result, inform research endeavors focused on preventing postpartum incontinence.

## Figures and Tables

**Figure 1 jcm-12-06037-f001:**
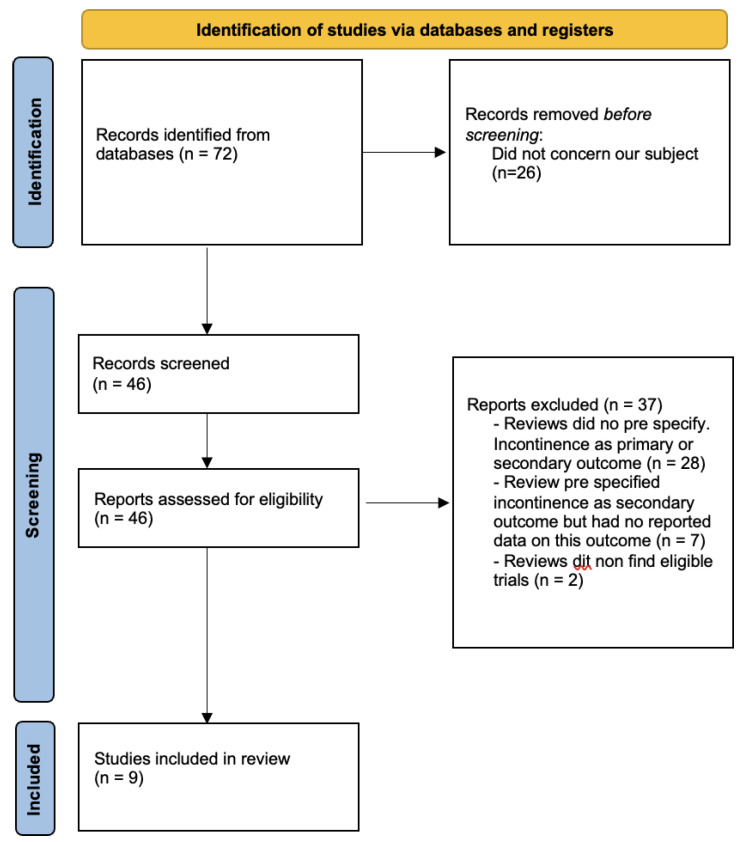
Review flow diagram.

**Table 1 jcm-12-06037-t001:** Characteristics of excluded reviews.

Review ID	Reason for Exclusion
Kettle et al.	No outcome focused on incontinence
Hodnett et al.	No outcome focused on incontinence
O’Kelly et al.	No outcome focused on incontinence
Bonet et al.	No outcome focused on incontinence
Liabsuetrakul et al.	No outcome focused on incontinence
Buppasiri et al.	No outcome focused on incontinence
Nygaard et al.	No outcome focused on incontinence
Shepherd et al.	No outcome focused on incontinence
Lavender et al.	Not concerning perineal trauma in postpartum
McIntyre et al.	Not concerning perineal trauma in postpartum
Berghella et al.	Not concerning perineal trauma in postpartum
Bamigboye et al.	No outcome focused on incontinence
Sheperd et al.	Not concerning perineal trauma in postpartum
Simmons et al.	Not concerning perineal trauma in postpartum
Quijano et al.	Not concerning perineal trauma in postpartum
Kettle et al.	No outcome focused on incontinence
Bohren et al.	Not concerning perineal trauma in postpartum
Ciapponi et al.	Not concerning perineal trauma in postpartum
Amorim Adegboye	Not concerning perineal trauma in postpartum
Tieu et al.	Not concerning perineal trauma in postpartum
Lavender et al.	Not concerning perineal trauma in postpartum
Reveiz et al.	No outcome focused on incontinence
Brown et al.	Not concerning perineal trauma in postpartum
Ostaszkiewicz et al.	Not concerning perineal trauma in postpartum
Cluett et al.	No outcome focused on incontinence
Middleton et al.	No outcome focused on incontinence
Boulvain et al.	No outcome focused on incontinence
Horey et al.	Not concerning perineal trauma in postpartum
Balogun et al.	No outcome focused on incontinence
Turawa et al.	No outcome focused on incontinence
Lawrie et al.	Not concerning perineal trauma in postpartum
Brown et al.	Not concerning perineal trauma in postpartum
East et al.	No outcome focused on incontinence
Walker et al.	Not concerning perineal trauma in postpartum
Sandall et al.	Not concerning perineal trauma in postpartum
Cody et al.	Not concerning perineal trauma in postpartum
Wuytack et al.	No outcome focused on incontinence
Costley et al.	No outcome focused on incontinence
Abalos et al.	No outcome focused on incontinence
Maeda et al.	Not concerning perineal trauma in postpartum
Hay-Smith et al.	Not concerning perineal trauma in postpartum
Bietsy et al.	No outcome focused on incontinence
Hedayati et al.	No outcome focused on incontinence
Bonet et al.	No outcome focused on incontinence
Basevi et al.	Not concerning perineal trauma in postpartum
Downe et al.	No outcome focused on incontinence
Tieu et al.	Not concerning perineal trauma in postpartum
Dudley et al.	No outcome focused on incontinence
Maher et al.	Not concerning perineal trauma in postpartum
Baessler et al.	Not concerning perineal trauma in postpartum
Hay smith	No outcome focused on incontinence
Hedayati et al.	No outcome focused on incontinence
Majoko et al.	No outcome focused on incontinence
Herbison et al.	Not concerning perineal trauma in postpartum
Torvaldsen et al.	Secondary outcomes included Incidence of urinary incontinence postpartum Incidence of faecal incontinence postpartum No outcome data for these outcomes
Elharmeel et al.	Secondary outcomes included Urinary and faecal incontinence No outcome data for theses outcomes
Dodd et al.	Secondary outcomes included Symptoms related to pelvic floor damage No outcome data for this outcome
Gupta et al.	Secondary outcome included Urinary or faecal incontinence No outcome data for this outcome
Aasheim et al.	Secondary outcome included -Stress incontinence No outcome data for these outcomes
Zhou F et al.	Secondary outcomes Urinary incontinence Faecal incontinence No outcome data for these outcomes
Hofmeyr et al.	Secondary outcomes included Faecal incontinence Urinary incontinence No outcome data for these outcomes
Farrar et al.	Primary outcome included Anal incontinence Secondary outcome included Faecal incontinence Urinary incontinence No eligible trials identified
Hofmeyr et al.	Secondary outcomes included Urinary incontinence Flatus incontinence Faecal incontinence No studies included

**Table 2 jcm-12-06037-t002:** Characteristics of included reviews.

Review ID	Date of Search	No. Included trials, Countries and Years of Publication No. Participants	Inclusion Criteria	Relevant Intervention and Comparison	Outcomes and Period of Evaluation
**Caesarean delivery for the prevention of anal incontinence** **Nelson 2010**	July 2009	21 studies, 1 RCT and 20 non-randomized studies 31,698 women	Women with a history of pregnancy and delivery of a live infant, including breech presentations and twin pregnancies.	Caesarean delivery versus vaginal delivery	Anal incontinence as primary outcome
**Choice of instruments for assisted vaginal delivery** **O’Mahony** **2010**	31 May 2010	32 RCTs 6597 women	Women in the second stage of labour due for instrumental vaginal delivery.	Any forceps vaginal birth; Any vacuum vaginal birth; Specific type of forceps vaginal birth; Specific type of vacuum vaginal birth.	Urinary and faecal incontinence as secondary outcomes
**Antenatal perineal massage for reducing perineal trauma** **Beckmann 2013**	23 October 2012	4 RCTs 2497 women	Pregnant women who are planning vaginal birth.	Any method of perineal massage undertaken by women and/or her partner versus no massage.	Urinary and faecal incontinence as secondary outcomes
**Methods of repair for obstetric anal sphincter injury** **Fernando 2013**	30 September 2013	6 RCTs 588 women	Women who sustained OASIS and in whom the repair was performed in the immediate postpartum period (primary repair).	Overlap versus end- to-end technique.	Anal incontinence as primary outcome
**Planned caesarean section for term breech delivery** **Hofmeyr 2015a**	31 March 2015	3 RCTs 2396 women	Women with breech presentation considered suitable for vaginal delivery.	Planned caesarean section compared with planned vaginal birth.	Urinary and faecal incontinence as secondary outcomes
**Use of endoanal ultrasound for reducing the risk of complications related to anal sphincter injury after vaginal birth** **Walsh 2015**	31 August 2015	1 RCT 752 women	Women after a vaginal birth, including spontaneous and assisted vaginal births.	Endo-anal ultrasound (EAUS) performed following vaginal birth and prior to repair of any perineal trauma versus no use of EAUS. EAUS performed following vaginal birth after repair of any perineal trauma, including women undergoing EAUS during subsequent pregnancies.	Severe anal incontinence at ≥ six months as primary outcome
**Planned caesarean section for women with a twin pregnancy** **Hofmeyr 2015b**	18 November 2015	2 RCTs 2864 women	Women with viable twin pregnancy considered suitable for vaginal birth.	Planned caesarean section compared with planned vaginal birth.	Urinary and faecal incontinence as secondary outcomes
**Selective versus routine use of episiotomy for vaginal birth** **Jiang 2017**	14 September 2016	12 RCTs 6177 women	Pregnant women having spontaneous or assisted vaginal births.	Performing episiotomy only if needed (’selective’ versus routine episiotomy).	Long-term effects including urinary incontinence and faecal incontinence as main outcomes
**Pelvic floor muscle training for preventing and treating urinary and faecal incontinence in antenatal and postnatal women** **Woodley 2020**	7 August 2019	46 RCTs 10,382 women	Populations included women who, at randomization, were continent (PFMT for prevention) or incontinent (PFMT for treatment), and a mixed population of women who were one or the other (PFMT for prevention or treatment).	Antenatal PFMT versus no PFMT, usual care or other control condition for the postnatal PFMT versus no PFMT, usual care.	Urinary and faecal incontinence as primary outcomes

**Table 3 jcm-12-06037-t003:** AMSTAR assessment for included reviews.

	Components of PICO	A priori Design	Explanation for Selection of Studies	Search Strategy	Duplicate Selection and Extraction	Excluded Studies List	Characteristics of Included Studies	Assessment Risk of Bias	Report of Fundings	Appropriate Methods for Résults in Meta-analysis	Assessment of RoB on the Results of the Meta-analysis	Account for RoB when Discussing Results of the Review	Explanation for any Heterogeneity	Investigation of Publication Bias and Discuss Its likely Impact of the Results	Report of any Conflict of Interest	Total
Fernando et al., 2013	+	+	+	+/−	+	+	+	+	−	+	+	+	+	−	+	High Quality
Woodley et al., 2020	+	+	−	+/−	+	+	+	+	+	+	+	+	+	−	+	High Quality
Jiang et al., 2017	+	+	−	+/−	+	+	+	+	+	+	+	+	+	−	−	High Quality
Walsh et al., 2015	+	+	+	+/−	+	/	+	+	+	/	/	+	+	/	+	High Quality
Nelson et al., 2010	+	+	+	+/−	+	+	+	+	−	+	+	+	+	−	+	High Quality
Hoffmeyr et al., 2015	+	+	+	+/−	+	+	+	+	+	+	+	+	+	−	+	High Quality
Hoffmeyr et al., 2015	+	+	+	+/−	+	+	+	+	+	+	+	+	+	−	+	High Quality
O’mahony et al., 2010	+	+	−	+/−	+	+	+/−	+	−	+	+	+	+	−	+	High Quality
Beckmann et al., 2013	+	+	−	+/−	+	+	+	+	−	+	+	+	+	−	+	High Quality

/: non applicable.

**Table 4 jcm-12-06037-t004:** ROBIS assessment.

	Study Eligibility Criteria	Identification and Selection of Studies	Data Collection and Study Appraisal	Synthesis and Findings	Overall Risk of Biais
“Caesarean delivery for the prevention of anal incontinence” Nelson et al., 2010	Low risk	Low risk	Low risk	Low risk	Low risk
“Choice of instruments for assisted vaginal delivery” O’Mahony et al., 2010	Low risk	Low risk	Low risk	Low risk	Low risk
“Antenatal perineal massage for reducing perineal trauma” Beckmann et al., 2013	Low risk	Low risk	Low risk	Low risk	Low risk
“Methods of repair for obstetric anal sphincter injury” Fernando 2013	Low risk	Low risk	Low risk	Low risk	Low risk
“Planned caesarean section for term breech delivery” Hofmeyr et al., 2015	Low risk	Low risk	Low risk	Low risk	Low risk
“Use of endoanal ultrasound for reducing the risk of complications related to anal sphincter injury after vaginal birth” Walsh 2015	Low risk	Low risk	Low risk	Low risk	Low risk
“Planned caesarean section for women with a twin pregnancy” Hofmeyr et al., 2015	Low risk	Low risk	Low risk	Low risk	Low risk
“Selective versus routine use of episiotomy for vaginal birth” Jiang 2017	Low risk	Low risk	Low risk	Low risk	Low risk
“Pelvic floor muscle training for preventing and treating urinary and faecal incontinence in antenatal and postnatal women” Woodley 2020	Low risk	Low risk	Low risk	Low risk	Low risk

**Table 5 jcm-12-06037-t005:** Summary of findings.

Intervention and Comparison	Outcome	Reference	RR Urinary Incontinence	RR Faecal Incontinence	Quality of the Evidence (GRADE)	Comments
Clinical examination versus use of endoanal ultrasound prior to repairing perineal tears.	Reduction in severe anal incontinence at greater than 6 months postpartum.	Walsh et al. [12]	/	RR 0.48 (95% CI 0.24 to 0.97)	Moderate	The use of endoanal ultrasound prior to repairing any perineal tears may reduce severe anal incontinence compared with routine care.
Selective versus routine use of episiotomy.	Long-term effects: urinary or faecal incontinence.	Jiang et al. [11]	RR 0.98 (95% CI 0.67 to 1.44)	No reported data	Low	Did not demonstrate a clear difference in urinary incontinence between selective or routine episiotomy at six months or more, postpartum.
Assess the effects of PFMT for preventing or treating urinary and faecal incontinence in postnatal women.	Self-reported urinary or faecal incontinence.	Woodley et al. [10]	**Early postnatal period:** RR 0.38 (95% CI 0.17 to 0.83) **Mid-postnatal period (three to six months):** RR0.71 (95% CI 0.54 to 0.95) **Late postnatal period (12 months):** RR 1.20 (95% CI 0.65 to 2.21)	No reported data	Low to moderate	PFMT may reduce urinary incontinence in early and mid-postnatal periods, but no difference was provided in late postnatal period.
Overlap repair versus end-to-end repair following OASIS.	Anal incontinence symptoms	Fernando et al. [13]	/	Anal incontinence at: **6 weeks**: RR 0.65 (95% CI 0.20 to 2.07) **3 months**: RR 0.84 (95% CI 0.06 to 12.73) **6 months**: RR 0.48 (95% CI 0.02 to 12.89) **12 months**: RR 0.37 (95% CI 0.03 to 4.68) **24 months**: RR 0.88 (95% CI 0.32 to 2.41) **36 months**: RR 1.01 (95% CI 0.34 to 2.98)	Low	No statistically significant difference in anal incontinence between the 2 repair techniques.
Caesarean delivery (CD) in comparison to vaginal delivery (VD) to preserve anal continence.	Anal incontinence	Nelson et al. [5]	/	CD vs. VD: OR 0.93 (95% CI 0.77 to 1.13)	Moderate	No benefit for CD over VD on anal incontinence has been demonstrated.
Effect of antenatal digital perineal massage on the incidence of perineal trauma at birth and Subsequent morbidity	Secondary outcomes: urinary or faecal incontinence at 3 months postpartum.	Beckmann et al. [9]	RR 0.90 (95% CI 0.74 to 1.08)	RR 0.70 (95% CI 0.27 to 1.80)	Moderate	No difference was seen in urinary or faecal incontinence after antenatal digital perineal massage.
Evaluate different instruments in terms of achieving a vaginal birth and avoiding significant morbidity for mother.	Secondary outcome at long-term: urinary and faeal incontinence.	O’Mahony et al [8]	/	Vacuum vs. Forceps: RR 0.56 95% CI 0.61 to 0.84, one trial, and 130 women	Moderate	Flatus incontinence/altered continence seems more likely with forceps.
Assess the effects of planned caesarean section for singleton breech presentation at term on measures of pregnancy outcome.	Secondary outcomes: urinary and faecal incontinence at 3 months and 2 years post- partum.	Hofmeyr et al. [6]	-At 3 months: RR 0.62 (95% CI 0.41 to 0.93) -at 2 years: RR 1.14 (95% CI 0.81 to 1.61)	-At 3 months: RR 0.54 (95% RR 0.18 to 1.62) -At 2 years: RR 1.11, (95% CI 0.47 to 2.58)	Low	At 3 months, planned caesarean may reduce urinary incontinence, but no difference detected for faecal incontinence. At 2 years, no difference was detected.
To determine the short- and long-term effects on mothers and their babies of planned caesarean section for twin pregnancy.	Secondary outcomes: urinary and faecal incontinence.	Hofmeyr et al. [7]	RR 0.87 (95% CI 0.64 to 1.18)	RR 1.02 (95% CI 0.69 to 1.51)	Moderate	No difference detected.

## Data Availability

No new data were created or analyzed in this study. Data sharing is not applicable to this article.

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
