# Peer review of "Antenatal, Intrapartum and Postpartum Interventions for Preventing Postpartum Urinary and Faecal Incontinence: An Umbrella Overview of Cochrane Systematic Reviews"

_jcm, 2023, doi:10.3390/jcm12186037_

Round 1

Reviewer 1 Report

Materials and methods

The authors should define the different terms of the inclusion periods used e.g. antenatal, intrapartum, postpartum, perpartum (?).

The terms used in the search strategy are wider than the terms in the aim of the overview, probably identifying more studies than necessary.  

The methods used to assess the quality of the included reviews are relevant and of high quality.

 References are lacking after the sentences ending at line 29, 224, 234, 367, 386, 400 (showing which reviews are based on RCTs and not).

Results

The structure throughout table 1 should be consequent. 

In table 3 it would help the reader if the name of the authors were presented in a first column.

Discussion

In general the discussion should include a more thorough discussion about similarities and differences between the reviews.

In line 359 the authors conclude that all the reviews have a low quality of evidence. What are this conclusion based on?

The discussion should also take into account the lacking information about the participant's continence status at inclusion. Are the interventions aimed at treatment or at prevention of urinary incontinence, fecal incontinence or both? 

From line 360 the authors discuss agreements and disagreements with other studies. This part should be better structured. In addition, it seems like relevant literature is lacking e.g. regarding pelvic floor muscle training. 

Authors' conclusions

The authors' conclusions is not correct according to the information earlier in the paper. Pelvic floor muscle training should be added as point 3) in the paragraph describing interventions with evidence showing reduction in urinary incontinence. 

Implications for research

Some general comments on possible problems related to the impact of Cochrane reviews, should be part of the paper. Have the authors reflected on the importance of the heterogeneity of the interventions within the some of the reviews? In several Cochrane reviews this introduce a large problem. E.g. when assessing the effect of interventions called "pelvic floor muscle training" the training protocols in the RCTs vary substantially. In addition, the number of participants in the RCTs included in the reviews, have a great impact on the results when  outcome data is pooled.  Low quality trials is a huge problem.

The implications for further research should be that all new trials must be planned and conducted in a way that secure high methodological quality. The RCT design is the gold standard design, but do not alone secure good trial quality. Inclusion / exclusion criteria must be clear, and participant numbers must follow power calculations. The new intervention tested must be compared to todays "best practice", outcome measures must follow international recommendations and so on. 

Regarding long-term follow up of interventions, it must be acknowledged that follow up in this period in women's lives is problematic.  Changes in the childbearing period with impact to the pelvic floor, as new pregnancies and deliveries, have relevance to the outcomes we aim to assess years after the intervention took place.

In general the authors should look closely to spelling (e.g. line 229, 233 ,234) and should be consequent in the use of either large or small letters in the beginning of the sentences (e.g. line 211-234).

Author Response

Thank you for your reviewing. Here are our answers (corrected in our review):

Materials and methods:

  • the antenatal period is during the pregnancy, postpartum is up to 3 years after the delivery.
  • indeed, we identified a lot of studies that were not related to our objective. We excluded them.
  • Thank you, we added the references.

Results

  • there is indeed a error in the uploading the table 3, we will send the right table for correction.

Discussion 

  • line 359: this conclusion is based on the grade approach
  • You're right, there is a lacking information about the patient's continence status at inclusion. The interventions aimed at prevention of urinary or fecal incontinence.

Authors' conclusion 

You're right, this is a mistake. Pelvic floor muscle training shows a potential reduction.

Implication for research 

About the heterogeneity, you're totally right. We added a comment in our text.

Reviewer 2 Report

Dear authors, congratulations for this good work. 

I believe it provides highly beneficial information for health professionals and for women. 

Only two issues seem to me that should be modified. Firstly, it would be necessary to justify why only one database is chosen and why that one in particular, and I consider this decision a limitation and as such it should be reflected and debated in the discussion. Secondly, the section on implications for practice leads me to confusion. On the one hand it states that there is moderate quality evidence that pelvic floor muscle training can reduce urinary incontinence in the early and mid postnatal period (but not the late period) but this is contradictory to what is stated in implications for practice where among the low quality evidence that showed no clear differences in urinary and fecal incontinence pelvic floor muscle training is included. por favor clarifique mejor las implicaciones para la práctica porque estas son sumamente importantes.  

Author Response

Thank you for your reviewing. 

For your first question: We chose to conduct our research exclusively within Cochrane because it is the most stringent database. But we are agree with you that having only one data base can be a limitation for our review. I added that in the discussion. 

For the second point, you're right, this is a mistake. Pelvic floor muscle training can reduce urinary incontinence in early et mid post partum. I have corrected that in the text. Thank you.

Round 2

Reviewer 1 Report

The paper has been improved sufficiently to be published.